# Nutritional Management Enhances the Recovery of Swallowing Ability in Older Patients with Sarcopenic Dysphagia

**DOI:** 10.3390/nu13020596

**Published:** 2021-02-11

**Authors:** Akio Shimizu, Ichiro Fujishima, Keisuke Maeda, Hidetaka Wakabayashi, Shinta Nishioka, Tomohisa Ohno, Akiko Nomoto, Jun Kayashita, Naoharu Mori

**Affiliations:** 1Department of Nutrition, Hamamatsu City Rehabilitation Hospital, Hamamatsu 433-8127, Japan; 2Department of Palliative and Supportive Medicine, Graduate School of Medicine, Aichi Medical University, Nagakute 480-1195, Japan; kskmaeda1701@gmail.com (K.M.); nmori@aichi-med-u.ac.jp (N.M.); 3Department of Rehabilitation Medicine, Hamamatsu City Rehabilitation Hospital, Hamamatsu 433-8127, Japan; ifujishima@sis.seirei.or.jp; 4Department of Geriatric Medicine, National Center for Geriatrics and Gerontology, Obu 474-8511, Japan; 5Department of Rehabilitation Medicine, Tokyo Women’s Medical University, Shinjuku-ku 162-8666, Japan; noventurenoglory@gmail.com; 6Department of Clinical Nutrition and Food Service, Nagasaki Rehabilitation Hospital, Nagasaki 850-0854, Japan; shintacks@yahoo.co.jp; 7Department of Dentistry, Hamamatsu City Rehabilitation Hospital, Hamamatsu 433-8127, Japan; tomohisa@zd5.so-net.ne.jp (T.O.); n.akiko-dent@sis.seirei.or.jp (A.N.); 8Department of Health Sciences, Faculty of Human Culture and Science, Prefectural University of Hiroshima, Hiroshima 734-8558, Japan; kayashita@pu-hiroshima.ac.jp

**Keywords:** nutritional intervention, older adults, oropharyngeal dysphagia

## Abstract

This study assessed whether a high provided energy of ≥30 kcal/ideal body weight (IBW)/day (kg) for patients with sarcopenic dysphagia effectively improved swallowing ability and the activities of daily living (ADLs). Among 110 patients with sarcopenic dysphagia (mean age, 84.9 ± 7.4 years) who were admitted to a post-acute hospital, swallowing ability and the ADLs were assessed using the Food Intake LEVEL Scale (FILS) and the Functional Independence Measure (FIM), respectively. The primary outcome was the FILS at discharge, while the secondary outcome was the achievement of the FIM with a minimal clinically important difference (MCID) at discharge. We created a homogeneous probability model without statistically significant differences using the inverse probability of treatment weighting (IPTW) method with and without a mean provided energy of ≥30 kcal/IBW/day (kg) for a period of 1 week of hospitalization and compared the outcomes between groups. A mean provided energy of ≥30 kcal/IBW/day (kg) was achieved in 62.7% of patients. In the IPTW model, the FILS and the rates of achieved MCID of the FIM at discharge were significantly higher in the mean provided energy of ≥30 kcal/IBW/day (kg) group (*p* = 0.004 and *p* < 0.001, respectively). A high provided energy for patients with sarcopenic dysphagia may improve swallowing ability and produce clinically meaningful functional outcomes.

## 1. Introduction

Dysphagia in older adults is a geriatric syndrome and is associated with malnutrition, dehydration, respiratory complications, aspiration pneumonia, and a poor prognosis in older patients [1,2]. Recently, sarcopenia, which is characterized by decreased muscle strength, physical function, and muscle mass and occurs in all muscles including the swallowing muscles, has been suggested to be an etiology of dysphagia [3]. Sarcopenic dysphagia is a pathology characterized by a decrease in the mass, strength, and function of the swallowing muscles [3,4].

Sarcopenic dysphagia is gaining increasing interest. In 2019, four academic societies published a position paper on the definition and diagnosis of sarcopenic dysphagia [3]. The current proposed diagnostic criteria include the presence of systemic sarcopenia and dysphagia, no apparent cause of dysphagia such as stroke or neurodegenerative disease, and the presence of low tongue strength [3,5]. The reported prevalence of sarcopenic dysphagia is 13–42% in older inpatients [6,7,8] and 32% in inpatients undergoing dysphagia rehabilitation [9]. The risk factors for the development of sarcopenic dysphagia are poor physical function, malnutrition, low muscle strength, and decreased muscle mass [6,7,8]. Furthermore, patients with sarcopenic dysphagia have poorer improvements in swallowing ability than patients with other causes of dysphagia [9].

Currently, evidence for therapeutic interventions for sarcopenic dysphagia is lacking [3]. Several case reports have shown that swallowing rehabilitation and interventions for systemic sarcopenia are also beneficial for sarcopenic dysphagia [10,11,12]. These case reports also implemented nutritional management with an energy intake of ≥30 kcal/ideal body weight (IBW)/day (kg) [10,11,12]. Physical rehabilitation is effective in improving swallowing ability [13,14]. Moreover, physical interventions and a high energy intake of ≥30 kcal/IBW/day (kg) improved tongue pressure in older patients with sarcopenia [14]. Tongue pressure is a key component of the diagnostic criteria for sarcopenic dysphagia [3,5]. These findings suggest that aggressive nutritional management [15] providing an energy intake of ≥30 kcal/IBW/kg/day for patients with sarcopenic dysphagia may improve functional outcomes, including swallowing ability and the activities of daily living (ADLs). However, the effects of nutritional management with a high provided energy in patients with sarcopenic dysphagia have not been clarified.

This study aimed to compare the effect of nutritional management using high versus low provided energy (≥30 kcal or <30 kcal/IBW/day (kg)) on changes in the improvement of swallowing ability and the ADLs for patients with sarcopenic dysphagia.

## 2. Materials and Methods

### 2.1. Participants and Study Design

This prospective cohort study, which was conducted in the post-acute care wards of the 225-bed Hamamatsu City Rehabilitation Hospital in Shizuoka, Japan, examined the eligibility of newly admitted patients who underwent post-acute rehabilitation between June 2019 and January 2020 for inclusion into the study. The inclusion criteria were age ≥65 years and the presence of sarcopenic dysphagia. We excluded patients without sarcopenia. Furthermore, patients who were discharged within 30 days because of a low likelihood of changes in swallowing ability and patients who had been transferred to an acute care hospital for the treatment of complicated acute diseases after hospitalization were excluded. The study was approved by the Ethics Committee of Hamamatsu City Rehabilitation Hospital (ID: 18–31), and written informed consent was obtained from all participants.

### 2.2. Diagnostic Criteria for Sarcopenic Dysphagia

The study complied with the currently proposed diagnostic criteria for sarcopenic dysphagia, and the patients underwent a diagnosis of the condition in a stepwise manner [3,5]. First, the presence of systemic sarcopenia was assessed. Second, the presence of dysphagia was evaluated to determine whether dysphagia was due to stroke or other neurologic problems. Sarcopenic dysphagia was diagnosed based on the presence of systemic sarcopenia and non-neurologic dysphagia. Finally, patients with low tongue strength were classified as having “probable sarcopenic dysphagia”, while those with unmeasurable or normal tongue strength were classified as having “possible sarcopenic dysphagia”. Sarcopenia was diagnosed based on low muscle strength and decreased muscle mass according to the Asian Working Group for Sarcopenia 2019 [16]. Handgrip strength (HGS) was measured using a Jamar Digital handgrip meter (MG-4800 digital grip strength meter; CHARDER Electronic, Taichung, Taiwan) with elbows bent at 90° and the left and right hands in the normal sitting position. A higher maximum value on either side was used in the analysis. Skeletal muscle mass index (SMI) was calculated by measuring appendicular skeletal muscle mass using bioimpedance analysis (BIA) (InBody S10; InBody Japan, Tokyo, Japan) and dividing by the height (m) squared. In this study, the cutoff values for low muscle mass were SMI < 7.00 kg/m^2^ for men and <5.70 kg/m^2^ for women. The cutoff values for low muscle strength were HGS < 28 kg for men and <18 kg for women [16]. Dysphagia was defined as a Food intake LEVEL Scale (FILS) [16] level of ≤8. The FILS is an ordinal scale that assesses the severity of dysphagia based on the daily diet consumed. Levels 1–3 relate to various degrees of non-oral feeding, levels 4–6 relate to various degrees of oral food intake and alternative nutrition, levels 7–8 relate to various degrees of oral food intake alone, level 9 relates to no dietary restriction but medical considerations given, and level 10 relates to normal oral food intake [17]. FILS was assessed by speech and language therapists. Tongue strength was measured using a tongue pressure measurement instrument (TPM-02, JMS, Hiroshima, Japan). The tongue strength of the participants was measured in a relaxed sitting position. The participants were instructed to place a balloon probe in their mouth, hold a plastic pipe in the center of the central incisors with their lips closed, raise their tongue, and press the probe against the hard palate with maximum effort. The values were recorded [18]. The highest of the three measurements was defined as the maximum tongue pressure (MTP). Low tongue pressure was defined as a tongue pressure of <20 kPa [3,5].

### 2.3. Determination of the Amount of Energy Provided and Calculation of Energy Intake

The amount of provided energy was determined by the attending physician based on the patient’s condition. In many cases, patients with higher amounts of provided energy were offered larger amounts of staple food and side dishes. A registered dietitian determined whether or not the amount of provided energy reached ≥30 kcal/IBW/day (kg). IBW was calculated using the following formula: IBW = height^2^ (m^2^) × 22 [19]. Moreover, this study considered the mean energy provided during the week after hospitalization. Several case reports have reported that energy provisions of ≥30 kcal/IBW (kg) during the nutritional intervention led to better outcomes for sarcopenic dysphagia [10,11,12]. We minimized the bias associated with determining the provided energy by using propensity scores to create an inverse probability of treatment weighting (IPTW) model. Mean energy intakes at 1 week during hospitalization and 1 week before discharge were investigated using the visual estimation method based on the food intake recorded by the ward nurses [20]. The registered dietitian calculated the energy intake based on this food intake record.

### 2.4. Main Outcome Measurements

The primary outcome was the FILS score at discharge. The secondary outcome was the achievement of the Functional Independence Measure (FIM) of the minimal clinically important difference (MCID) at discharge. The FIM is an indicator of the ADLs and consists of a motor domain of 13 subitems (motor FIM) and a cognitive domain of five subitems (cognitive FIM) [21]. The scores are evaluated on a seven-point ordered scale, ranging from total assistance to complete independence. The total FIM score ranges from 18 to 126, with lower scores indicating more dependency. In this study, the FIM was assessed by physical or occupational therapists. An FIM gain of ≥22 points was defined as achieving the FIM of the MCID, as previously described [22]. The FIM gain was calculated by subtracting the FIM at admission from the FIM at discharge.

### 2.5. Other Variables

Data on other variables including sex, age, presence of primary diseases, comorbidity, duration from symptom onset to admission, body mass index (BMI), nutritional status, and cognitive function were collected. Comorbidity was assessed using the Charlson Comorbidity Index (CCI) [23]. The CCI was scored for 19 diseases that affect mortality, with higher scores indicating severe comorbidities. BMI was calculated by dividing the body weight (kg) by the square of the height (m). Nutritional status was assessed by a registered dietitian using the Mini Nutritional Assessment Short Form (MNA-SF) [24]. The MNA-SF scores range from 0 to 14, with lower scores indicating poorer nutritional status. Cognitive function was assessed by occupational therapists using the Mini-Mental State Examination (MMSE) [25]. The MMSE score ranges from 0 to 30, with lower scores indicating poorer cognitive function.

### 2.6. Statistical Analyses

We created an IPTW model [26,27] using propensity scores to control and adjust for patient characteristics and confounders during hospitalization. In this model, we assessed whether the baseline variables were unbiased according to the effect sizes and *p*-values. The propensity score used to weigh the IPTW model was the predictive probability of grouping calculated for each case using a logistic regression model with 14 independent variables (age, sex, time from symptom onset to admission, underlying disease, CCI, BMI, MNA-SF, MMSE, HGS, SMI, FIM, FILS, MTP, and possible sarcopenic dysphagia) determined on admission. The outcomes were compared between patients with a mean energy of ≥30 kcal/IBW/day (kg) and those with <30 kcal/IBW/day (kg) during the 1 week of hospitalization. Data of continuous and ordinal variables were presented as mean ± standard deviation and median (25th–75th percentiles), respectively. The differences were analyzed using Student’s t- or Mann–Whitney U tests. Categorical variables were expressed as percentages, and the differences were analyzed using chi-square tests. For the parametric t- and nonparametric tests, the effects were reported using Cohen’s d and r, respectively. The chi-square test effect was reported using Cramer’s V. For all effect sizes, values of 0.10–0.20, 0.20–0.30, >0.30, and <0.10 were considered weak, midrange, large, and none, respectively [28]. Weighted univariate logistic regression analyses were performed to determine the effect of mean energy on the improvement of swallowing ability and the MCID of the FIM. All statistical analyses were performed using IBM SPSS Statistics for Windows, version 23.0 (IBM Japan, Tokyo, Japan) package. *p*-values < 0.05 were considered statistically significant.

## 3. Results

During the study period, 129 patients with sarcopenic dysphagia were enrolled. Of them, 12 were discharged within 30 days and seven were transferred to acute care hospitals and finally excluded from the study. A total of 110 patients with sarcopenic dysphagia (mean age, 84.9 ± 7.4 years) were included during the observation period (Figure 1).

Participants’ characteristics are listed in Table 1. A mean provided energy of ≥30 kcal/IBW/day (kg) was achieved in 62.7% of the patients. In addition, a mean energy of ≥35 kcal was observed in only 7.3% of patients. The ≥30 kcal/IBW/day (kg) group showed a lower MNA-SF score than the <30 kcal/IBW/day (kg) group (*p* = 0.016). While not reaching statistical significance, when focusing on effect size, the <30 kcal/IBW/day (kg) group appeared to have a lower BMI and more subjects had a higher HGS and SMI.

However, the inverse probability of treatment weighting model constructed from all cases showed no significant differences between the groups for any parameter determined at hospitalization (Table 2).

The comparison of variables at discharge (Table 3) showed that the primary outcome, FILS at discharge, was significantly higher (*p* = 0.004) and had a small effect size (−0.20) in the ≥30 kcal/IBW/day (kg) group than in the <30 kcal/IBW/day (kg) group. Similarly, the secondary outcome, achieving the MCID of the FIM, was significantly higher (*p* < 0.001) in the ≥30 kcal/IBW/day (kg) group than in the <30 kcal/IBW/day (kg) group, with a small effect size (0.26). The mean energy intake before discharge was significantly higher in the ≥30 kcal/IBW/day (kg) group than in the <30 kcal/IBW/day (kg) group (effect size 0.85 and *p* < 0.001).

In the weighted univariate regression analysis, the mean provided energy of ≥30 kcal/IBW/day (kg) did not reach statistical significance in improving the FILS ≥ 1 point at discharge (odds ratio (OR), 1.664; 95% confidence interval (CI): 0.657–4.215, *p* = 0.283) but was statistically significant in improving the FILS ≥ 2 points (OR: 4.958; 95% CI: 1.044–23.543, *p* = 0.044). Additionally, the mean provided energy of ≥30 kcal/IBW/day (kg) reached statistical significance in achieving the MCID of the FIM (OR: 3.032; 95% CI: 1.126–8.167, *p* = 0.028) (Figure 2).

## 4. Discussion

The present study, conducted in older patients with dysphagia undergoing post-acute rehabilitation, showed that the higher the amount of provided energy, the more significant the improvement in swallowing ability and the ADLs.

In patients with sarcopenic dysphagia undergoing physical rehabilitation, including swallowing rehabilitation, nutritional management with a high provided energy led to a significant improvement in swallowing status (FILS improvement of ≥2 points). A meta-analysis of randomized controlled trials conducted in patients with sarcopenia showed that nutritional management was effective in improving whole-body muscle strength [29]. Some case reports of patients with sarcopenic dysphagia reported that an IBW-based nutritional management providing energy of ≥30 kcal/day (kg) was effective in improving swallowing ability [10,11,12]. In short, nutritional management may be effective in improving muscle strength in patients with systemic sarcopenia. Physical interventions may also affect muscle tissue throughout the body, including swallowing-related muscle groups [30,31]. Physical rehabilitation may be effective in improving swallowing ability [13,14]. Nagano et al. [14] showed that physical rehabilitation was associated with a significant increase in tongue strength, a swallowing-related muscle group. A combination of nutritional management and physical interventions is considered preferable for the treatment of systemic sarcopenia [32]. Similarly, in patients with sarcopenic dysphagia, a combination of nutritional management and physical interventions may improve systemic muscle strength and function, including those of the swallowing muscles. In the present study, application of the combination of physical rehabilitation and nutritional management in patients with sarcopenic dysphagia may have resulted in significant improvements in swallowing ability.

Nutritional management with a high provided energy was effective in improving the ADLs in patients with sarcopenia. Nutrient intake reportedly affects the improvement of the ADLs in hospitalized older patients [33,34,35]. A high energy intake in post-acute wards improved the physical and functional outcomes in hospitalized older adults [33]. A high energy provision in the ≥30 kcal/IBW/day (kg) group was presumably associated with an increased nutrient intake and led to a better improvement in the ADLs. Protein provision is also thought to be an effective nutritional management strategy for systemic sarcopenia [36]. A protein intake of 1.2 g/IBW/day (kg) may effectively improve tongue strength in older adults with sarcopenia [14]. Further research is needed to determine the effectiveness of the protein provided in the nutritional management of patients with sarcopenic dysphagia.

Nutritional management that provides an energy level of ≥30 kcal/day (kg) based on the IBW in patients with sarcopenic dysphagia may aid in higher compliance rates and be feasible in the clinical setting. In the present study, approximately 60% of patients achieved an energy level of ≥30 kcal/IBW/day (kg). Nishiyama et al. [34] reported that approximately 50% of stroke patients achieved a mean energy intake of ≥26 kcal/IBW/day (kg). The group that had provided an energy level of ≥30 kcal/IBW/day (kg) in this study also had a mean energy intake of approximately 26 kcal/IBW/day (kg), with a higher compliance rate than that reported in the previous study. Therefore, nutritional management providing an energy level of ≥30 kcal/IBW/day (kg) may show high compliance rates. However, nutritional management with more provided energy may be necessary for patients with sarcopenic dysphagia, who have a poorer nutritional status. The <30 kcal/IBW/day (kg) group had a significantly lower baseline MNA-SF score before being weighted using the IPTW method. These patients with a poorer nutritional status on admission may have potential anorexia, low activity, or other problems. Individualized nutritional management is effective in improving their nutritional status and ADLs [37,38]. Considering these findings, individualized and nutritional management with more provided energy is important to increase compliance with the 30 kcal/IBW/day (kg) intake. In some patients, oral intake may be inadequate even with individualized nutritional management. In the future, it is necessary to establish an effective nutritional management for these patients. The position paper on sarcopenic dysphagia suggests a provided energy of ≥35 kcal/IBW/day (kg) in nutritional management [3]. However, few participants in the current study reached a provided energy of ≥35 kcal/IBW/day (kg). Nutritional management with a high provided energy of ≥30 kcal/IBW/day (kg) may be more feasible. Moreover, energy intakes at hospitalization and discharge were significantly higher in the ≥30 kcal/IBW/day (kg) group. A high energy provision based on IBW from the time of hospitalization may provide patients with an opportunity for increased energy intake. Similarly, the ≥30 kcal/IBW/day (kg) group may have consumed a larger amount of diet than the <30 kcal/IBW/day (kg) group. Therefore, a larger amount of diet intake may increase the swallowing frequency and positively affect the swallowing muscles.

This study has several limitations. First, this was a prospective observational study; a randomized controlled trial would have been preferable to better define the effect of nutritional management with a high provided energy for sarcopenic dysphagia. However, in this study, we created an IPTW model for the factor of modifiable provided energy that used variables known before the intervention and adjusted to minimize potential bias. Second, this was a single-center study, which may limit the generalizability of the results. Multicenter intervention studies are needed to assess the effects of nutritional management with a high provided energy in patients with sarcopenic dysphagia. Lastly, it is possible that the patients excluded from this study may be refractory to nutritional management based on ≥30 kcal/IBW/day (kg). Multicenter intervention studies are needed to assess the effects of nutritional management with a high provided energy in patients with sarcopenic dysphagia. Therefore, we cannot deny the possibility of the presence of selection bias among the participants.

## 5. Conclusions

The results of the current study showed that nutritional management based on ≥30 kcal/IBW/day (kg) in older hospitalized patients with sarcopenic dysphagia may significantly improve swallowing ability and provide clinically meaningful improvements in the ADLs. Nutritional management with a high provided energy may be desirable for better clinical outcomes in patients with sarcopenic dysphagia.

## Figures and Tables

**Figure 1 nutrients-13-00596-f001:**
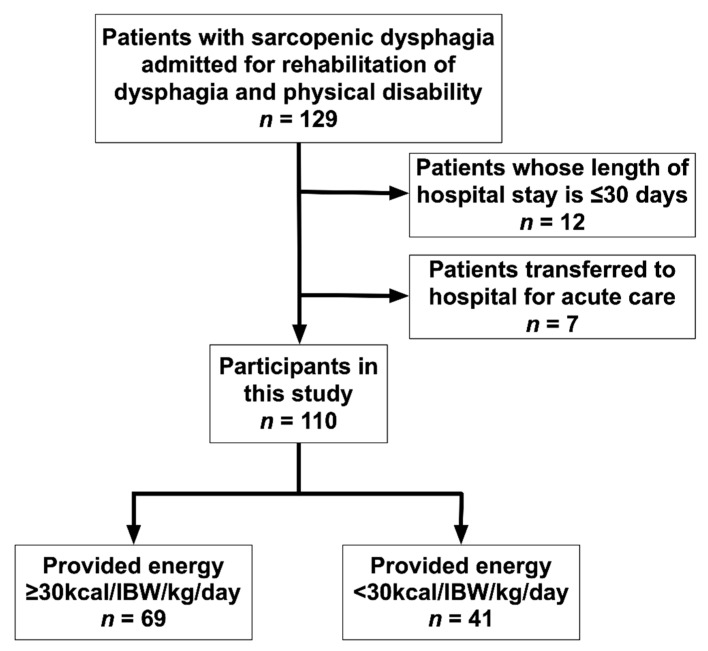
Flowchart of the study participant selection process. Patients with sarcopenic dysphagia aged ≥65 years (*n* = 129) were examined for participation eligibility. Finally, 110 patients were included in the study.

**Figure 2 nutrients-13-00596-f002:**
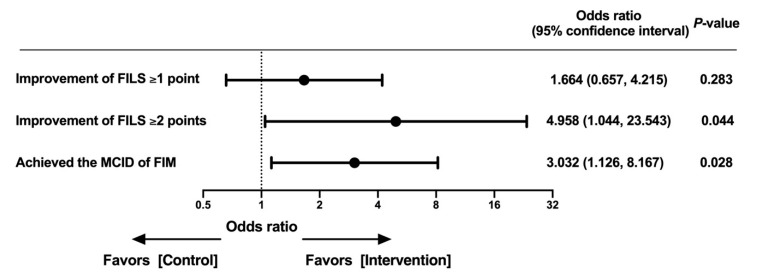
Effect of the mean provided energy on the outcomes at discharge. Effect of the mean provided energy during one week of hospitalization on improvement in swallowing ability at discharge and achievement of the MCID of the FIM. Abbreviations: FILS: Food Intake Level Scale, MCID: minimal clinically important difference, FIM: Functional Independence Measure, IBW: ideal body weight.

**Table 1 nutrients-13-00596-t001:** Comparison of baseline variables in the crude population.

*n* = 110	Amount of Energy Provided during 1 Week of Hospitalization		
	≥30 kcal/IBW/Day (kg)	<30 kcal/IBW/Day (kg)	ES (*p*-Value)
Participants, *n* (%)	69 (62.7)	41 (37.3)	
Age, years	84.9 ± 7.1	84.9 ± 8.1	0.00 (0.979)
Female, *n* (%)	28 (67.6)	5 (79.5)	0.12 (0.178)
Primary disease, *n* (%)			0.10 (0.736)
-Hip fracture	46 (62.2)	28 (69.9)	
-Compression fracture	5 (2.7)	5 (12.3)	
-Other fractures	8 (13.5)	8 (8.2)	
-Deconditioning	10 (21.6)	10 (9.6)	
Onset-to-admission duration, days	20 (14.5–33.5)	24 (19–31)	−0.10 (0.319)
Charlson Comorbidity Index, points	0 (0–2)	2 (0–3)	−0.16 (0.105)
Body mass index, kg/m^2^	19.8 ± 3.0	18.9 ± 2.8	0.31 (0.114)
Handgrip strength, kg	12.1 ± 4.9	13.2 ± 6.1	0.20 (0.324)
Skeletal muscle mass index, kg/m^2^	4.7 ± 0.8	4.9 ± 1.0	0.23 (0.352)
Mini Nutritional Assessment-Short Form, points	6 (5–8)	5 (3–7)	−0.23 (0.016)
Mini-Mental State Examination, points	22 (15.5–26)	24 (16–26.5)	−0.10 (0.304)
Functional Independence Measure, points	75 (64.5–81)	72 (54–90.5)	−0.01 (0.958)
Food Intake LEVEL Scale, points	8 (7,8)	8 (7,8)	−0.03 (0.719)
Maximum tongue pressure, kPa	18.4 ± 9.1	17.2 ± 9.2	0.13 (0.515)
Possible sarcopenic dysphagia, *n* (%)	40 (58.0)	24 (58.5)	0.00 (0.954)
Mean energy intake during one week of hospitalization/IBW (kg)	26.3 ± 7.2	20.5 ± 5.3	0.88 (<0.001)

Abbreviations: IBW: ideal body weight, ES: effect size.

**Table 2 nutrients-13-00596-t002:** Comparison of baseline variables in the inverse probability of treatment weighting (IPTW) model.

*n* = 219	Amount of Energy Provided during 1 Week of Hospitalization		
	≥30 kcal/IBW/Day (kg)	<30 kcal/IBW/Day (kg)	ES (*p*-Value)
Participants, *n* (%)	113 (51.6)	106 (48.4)	
Age, years	85.5 ± 6.9	85.7 ± 8.1	0.01 (0.934)
Female, *n* (%)	78 (69.0)	74 (69.8)	0.00 (0.900)
Primary disease, *n* (%)			0.05 (0.862)
-Hip fracture	78 (69.0)	72 (67.9)	
-Compression fracture	10 (8.8)	12 (11.3)	
-Other fractures	10 (8.8)	7 (6.6)	
-Deconditioning	15 (13.3)	15 (14.2)	
Onset-admission duration, days	24 (16–33)	24 (18–30)	−0.03 (0.638)
Charlson Comorbidity Index, points	2 (0–2)	1 (0–2)	−0.04 (0.542)
Body mass index, kg/m^2^	19.5 ± 2.8	19.8 ± 3.0	0.10 (0.481)
Handgrip strength, kg	13.0 ± 5.2	12.7 ± 5.6	0.06 (0.672)
Skeletal muscle mass index, kg/m^2^	4.8 ± 0.8	4.8 ± 1.0	0.01 (0.892)
Mini Nutritional Assessment Short Form, points	6 (5–7)	5 (4–9)	−0.01 (0914)
Mini-Mental State Examination, points	23 (17–27)	24 (16–25)	−0.01 (0.930)
Functional Independence Measure, points	73 (65–81]	73 (55–91)	−0.02 (0.780)
Food Intake LEVEL Scale, points	8 (8–8)	8 (7,8)	−0.07 (0.290)
Maximum tongue pressure, kPa	18.5 ± 8.4	19.1 ± 9.6	0.07 (0.572)
Possible sarcopenic dysphagia, *n* (%)	61 (54.0)	52 (49.0)	0.04 (0.466)
Mean energy intake during one week of hospitalization/IBW (kg)	26.4 ± 6.7	20.7 ± 5.3	0.94 (<0.001)

Abbreviations: IPTW, inverse probability of treatment weighting; IBW, ideal body weight; ES, effect size.

**Table 3 nutrients-13-00596-t003:** Comparison of the variables at discharge in the IPTW model.

*n* = 219	Amount of Energy Provided during 1 Week of Hospitalization		
	≥30 kcal/IBW/Day (kg)	<30 kcal/IBW/Day (kg)	ES (*p*-Value)
Length of hospital stay, days	57 (40–73)	59 (42–78)	−0.09 (0.189)
Duration of rehabilitation, units/day	5.8 ± 1.4	5.6 ± 1.6	0.13 (0.233)
Body mass index, kg/m^2^	19.7 ± 2.9	19.2 ± 3.5	0.16 (0.343)
Handgrip strength, kg	14.6 ± 5.6	13.9 ± 5.4	0.13 (0.308)
Skeletal muscle mass index, kg/m^2^	5.0 ± 0.9	4.9 ± 1.0	0.11 (0.361)
Functional Independence Measure, points	101 (80–110)	95 (78–111)	−0.08 (0.237)
Food Intake LEVEL Scale, points	8 (8–9)	8 (8–8)	−0.20 (0.004)
Improvement of Food Intake LEVEL Scale ≥ 1 points, *n* (%)	56 (49.6)	39 (36.8)	0.12 (0.065)
Improvement of Food Intake LEVEL Scale ≥ 2 points, *n* (%)	28 (24.8)	7 (6.6)	0.25 (<0.001)
Maximum tongue pressure, kPa	20.4 ± 8.1	20.1 ± 10.2	0.03 (0.804)
Functional Independence Measure gain	23.2 ± 12.7	18.9 ± 12.4	0.34 (0.013)
Achieved the MCID of Functional Independence Measure, *n* (%)	59 (52.2)	28 (26.4)	0.26 (<0.001)
Mean energy intake before one week at discharge/IBW (kg)	30.3 ± 5.7	24.9 ± 7.0	0.85 (<0.001)

Abbreviations: IPTW, inverse probability of treatment weighting; IBW, ideal body weight; ES, effect size; MCID, minimal clinically important difference.

## Data Availability

Data sharing is not applicable to this article.

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
