# Peer review of "Nutritional Management Enhances the Recovery of Swallowing Ability in Older Patients with Sarcopenic Dysphagia"

_nutrients, 2021, doi:10.3390/nu13020596_

Round 1

Reviewer 1 Report

Thank you for the opportunity for reviewing this manuscript. This is an interesting topic of a single group observational study assessing the aggressive nutritional management effects on swallowing ability and activities of daily living on patients with sarcopenic dysphagia in comparison to an intake of <30kcal/IBM/day. The manuscript is clearly written and the authors used validated and appropriate assessment tools. However, the objective of the study should be better defined in accordance to the methodology, the discussion merits further work and some limitations should be recognized. Nevertheless, I think that the study has clinical interest and that is why I would recommend a major revision of the manuscript.

Line 72-74:Given that this is a single group observational study, I would not determine the aim of the study in terms of effectiveness.

I suggest to include the comparation of both nutritional managements effects on the assessed parameters

Line 77-86:I am not sure if excluding those participants from the study may constitute a selection bias. How it is posible to determine whether the assessed intervention is effective for improving swallowing ability if participants that dont´n show clear improvements are not included? This should be recognized as a limitation of the study

Line 100-107:Given that both right and left handgrip were measured, it should be specified which one is going to be used for screening for low muscle strength

Line 122-125:I´m not sure of have seen these results; maybe Authors mean during hospitalization?

Line 154-174:I´m sorry but I am not familiarized with this statistical methodology and I am not able to do any comments on it

Line 180-185: It seems that this lower score on the MNA-SF could explain the aggresive nutritional management on the group that received more than 30kcal/IBM/day. Please, commemt on that in the discussion section

Line 234-238:I wonder if an aggresive nutritional management could be also understood as an intensive training of the swallowing muscles; in fact, it seems to me that a greater effort for those muscles might be asked to the patients when eating more than 30kcal/IBM/day in comparison with less than 30kcal/IBM/day. Could authors give more information regarding what kind of food was given to participants? I assume that all participants eaten the same food but in different amounts; Authors should provide this information in the method section 

Line 273-276:This conclusion should be modified according to the objectives and limitation of the study; 

Author Response

Reviewer 1.

Thank you for your positive comments in the previous review, which had helped us improve the quality of our manuscript. We have made additional changes in this revision based on your comments. Our changes have been marked in red font in the revised manuscript.

Comment #1: Line 72-74:Given that this is a single group observational study, I would not determine the aim of the study in terms of effectiveness.

I suggest to include the comparation of both nutritional managements effects on the assessed parameters

Response: Thank you for pointing this out. We agree with the reviewer’s comment. This study was a single-group observational study. Therefore, it was not possible to directly assess whether or not aggressive nutritional management was effective. As suggested by the reviewer, it was considered a more appropriate objective to clarify whether nutritional management based on the amount of provided energy (≥30 kcal or <30 kcal/IBW/day (kg)) would affect the endpoints. Therefore, we have modified the relevant sentences in the Introduction section as follows:

This study aimed to compare the effect of nutritional management high and low provided energy (≥ 30 kcal or < 30 kcal/IBW/day [kg] ) on changes in the improvement of swallowing ability and ADLs for patients with sarcopenic dysphagia.” (Lines 72-74)

and

“The present study conducted in older patients with dysphagia undergoing post-acute rehabilitation showed that the higher the amount of provided energy, the significant was the improvement in swallowing ability and ADLs.” (Lines 215-217)

Comment #2: Lines 77–86:I am not sure if excluding those participants from the study may constitute a selection bias. How is it possible to determine whether the assessed intervention is effective for improving swallowing ability if participants that dont´n show clear improvements are not included? This should be recognized as a limitation of the study.

Response: We agree with the reviewer’s comment. , we have added the following sentences in the Discussion section:

“Lastly, it is possible that the patients excluded from this study may be refractory to aggressive nutritional management. Therefore, we cannot deny the possibility of the presence of selection bias among participants.” (Lines 277-279)

Comment #3: Given that both right and left handgrip were measured, it should be specified which one is going to be used for screening for low muscle strength.

Response: Thank you for your comment. We used the maximum handgrip strength of either the left or right side in our study. Therefore, we have modified the following sentences in the Discussion and Reference sections:

“The higher maximum value on either side was used in the analysis.” (Line 99)

Comment #4: Line 122-125:I´m not sure of have seen these results; maybe Authors mean during hospitalization?

Response: We thank the reviewer for providing these insights. In previous case reports, energy of up to ≥30 kcal/IBW/day (kg) provided during nutritional interventions affected the endpoints. Therefore, we have modified the following sentences in the Methods section:

“Several case reports have reported that energy provisions of ≥30 kcal/IBW (kg) during the nutritional intervention led to better outcomes for sarcopenic dysphagia.” (Lines 123-124)

Comment #5: Line 154-174:I´m sorry but I am not familiarized with this statistical methodology and I am not able to do any comments on it.

Response: Thank you for your comment. In this study, the patients were classified into two groups: ≥30 kcal/IBW/day (kg) or <30 kcal/IBW/day (kg). However, because this was an observational study, these two groups may differ in terms of patients' background characteristics. Therefore, we used a propensity score to calculate the probability of patients being classified into these two groups. The inverse probability of treatment weighting (IPTW) method was then used to adjust the background to minimize the difference between them statistically.

Comment #6: Line 180-185: It seems that this lower score on the MNA-SF could explain the aggresive nutritional management on the group that received more than 30kcal/IBM/day. Please, commemt on that in the discussion section.

Response: We agree with the reviewer’s comment. Poorer nutritional status may have potential factors such as anorexia and decreased activity. Therefore, more aggressive nutritional management may be needed in patients with sarcopenic dysphagia who have a lower MNA-SF. We have added the following sentences in the Discussion and Reference sections:

“However, more aggressive nutritional management may be necessary for patients with sarcopenic dysphagia, who have a poorer nutritional status. The <30 kcal/IBW/day (kg) group had a significantly lower baseline MNA-SF score before being weighted using the IPTW method. These patients with a poorer nutritional status on admission may have potential anorexia, low activity, or other problems. Individualized nutritional management is effective in improving their nutritional status and ADLs.[37,38] Considering these findings, individualized and more aggressive nutritional management is important to increase compliance with the 30 kcal/IBW/day (kg) intake. In some patients, oral intake may be inadequate even with individualized nutritional management. In the future, it is necessary to establish effective nutritional management for these patients.” (Lines 253-261)

and

  1. Schuetz, P.; Fehr, R.; Baechli, V.; Geiser, M.; Deiss, M.; Gomes, F.; Kutz, A.; Tribolet, P.; Bregenzer, T.; Braun, N.; et al Individualised nutritional support in medical inpatients at nutritional risk: A randomised clinical trial. Lancet 2019, 393, 2312–2321.
  2. Shimazu, S.; Yoshimura, Y.; Kudo, M.; Nagano, F.; Bise, T.; Shiraishi, A.; Sunahara, T. Frequent and personalized nutritional support leads to improved nutritional status, activities of daily living, and dysphagia after stroke. Nutrition 2020, 83, 111091.

Comment #7: Line 234-238:I wonder if an aggresive nutritional management could be also understood as an intensive training of the swallowing muscles; in fact, it seems to me that a greater effort for those muscles might be asked to the patients when eating more than 30kcal/IBM/day in comparison with less than 30kcal/IBM/day. Could authors provide more information regarding what kind of food was given to participants? I assume that all participants eaten the same food but in different amounts; Authors should provide this information in the method section.

Response: We thank the reviewer for providing these insights. In our study, we provided food textures that were appropriate for the patients' swallowing ability. Therefore, we did not provide significantly different diets for each group. However, the nutrient densities may have differed between different food textures, such as pureed and regular. As pointed out by the reviewer, when energy was high, the amount of meals served increased. It is also possible that the higher amount of meals served may have led to increased opportunities for swallowing. Therefore, we have added the following sentences in the Methods and Discussion sections:

“In many cases, patients with higher amounts of provided energy were offered larger amounts of staple food and side dishes.” (Lines 119-120)

and

Similarly, the ≥30 kcal/IBW/day (kg) group may have consumed a larger amount of diet than the <30 kcal/IBW/day (kg) group. Therefore, a larger amount of diet intake may increase the swallowing frequency and positively affect the swallowing muscles.” (Lines 267-270)

Comment #8: Line 273-276:This conclusion should be modified according to the objectives and limitation of the study.

Response: We agree with the reviewer’s comment. Our study was an observational study; hence, we could not determine the effect of aggressive nutritional management practices on improved outcomes. Therefore, we have modified the following sentences in the Conclusions sections:

“The results of the current study showed that nutritional management based on ≥30 kcal/IBW/day (kg) in older hospitalized patients with sarcopenic dysphagia may significantly improve swallowing ability and clinically meaningful ADLs.” (Lines 281-283)

Reviewer 2 Report

This is an interesting study about a topic that must be recognized and investigated due to the clinical impact of this. 

I have some advices that the authors can do: 

  • You must include a flow chart of study groups. 
  • The group with <30 kcal/día had less intake, you should explain it in discussion. 

Author Response

Reviewer 2.

Thank you for your positive comments in the previous review, which helped us improve our manuscript. We have made additional changes in this revision based on the comments. Our changes have been marked in red font in the revised manuscript.

Comment #1: You must include a flow chart of study groups.

Response: We agree with the reviewer’s comment. As pointed out by the reviewer, presenting a flowchart of this study would better clarify details about the study participants. Therefore, we have added Figure 1.

Comment #2: The group with <30 kcal/día had less intake, you should explain it in discussion.

Response: We agree with the reviewer’s comment. Taking into account why the <30 kcal/IBW/day (kg) group had a low energy intake would be useful for effective nutritional management in the future. Therefore, we have added the following sentences in the Discussion and Reference sections:

“However, more aggressive nutritional management may be necessary for patients with sarcopenic dysphagia, who have a poorer nutritional status. The <30 kcal/IBW/day (kg) group had a significantly lower baseline MNA-SF score before being weighted using the IPTW method. These patients with a poorer nutritional status on admission may have potential anorexia, low activity, or other problems. Individualized nutritional management is effective in improving the nutritional status and ADLs. [37,38] Considering these findings, individualized and more aggressive nutritional management is important to increase compliance with the 30 kcal/IBW/day (kg) intake. In some patients, oral intake may be inadequate even with individualized nutritional management. In the future, it is necessary to establish effective nutritional management for these patients.” (Lines 253-261)

37. Schuetz, P.; Fehr, R.; Baechli, V.; Geiser, M.; Deiss, M.; Gomes, F.; Kutz, A.; Tribolet, P.; Bregenzer, T.; Braun, N.; et al Individualised nutritional support in medical inpatients at nutritional risk: A randomised clinical trial. Lancet 2019, 393, 2312–2321.

38.  Shimazu, S.; Yoshimura, Y.; Kudo, M.; Nagano, F.; Bise, T.; Shiraishi, A.; Sunahara, T. Frequent and personalized nutritional support leads to improved nutritional status, activities of daily living, and dysphagia after stroke. Nutrition 2020, 83, 111091.
